# Profile of a Multivariate Observation under Destructive Sampling—A Monte Carlo Approach to a Case of Spina Bifida

**DOI:** 10.3390/bioengineering11030249

**Published:** 2024-03-03

**Authors:** Tianyuan Guan, Rigwed Tatu, Koffi Wima, Marc Oria, Jose L. Peiro, Chia-Ying Lin, Marepalli. B. Rao

**Affiliations:** 1College of Public Health, Kent State University, Kent, OH 44242, USA; 2Division of Biostatistics and Bioinformatics, University of Cincinnati, Cincinnati, OH 45221, USA; 3The Center for Fetal and Placental Research, Pediatric General and Thoracic Surgery Division, Cincinnati Children’s Hospital Medical Center, Cincinnati, OH 45229, USAoriaalmc@ucmail.uc.edu (M.O.);; 4Orthopedic Surgery, University of Cincinnati College of Medicine, Cincinnati, OH 45267, USA; linc9@ucmail.uc.edu

**Keywords:** birth defects, hybrid polymer patches, destructive sampling, multivariate normal distribution

## Abstract

A biodegradable hybrid polymer patch was invented at the University of Cincinnati to cover gaps on the skin over the spinal column of a growing fetus, characterized by the medical condition spina bifida. The inserted patch faces amniotic fluid (AF) on one side and cerebrospinal fluid on the other side. The goal is to provide a profile of the roughness of a patch over time at 0, 4, 8, 12, and 16 weeks with a 95% confidence band. The patch is soaked in a test tube filled with either amniotic fluid (AF) or phosphate-buffered saline (PBS) in the lab. If roughness is measured at any time point for a patch, the patch is destroyed. Thus, it is impossible to measure roughness at all weeks of interest for any patch. It is important to assess the roughness of a patch because the rougher the patch is, the faster the skin grows under the patch. We use a model-based approach with Monte Carlo simulations to estimate the profile over time with a 95% confidence band. The roughness profiles are similar with both liquids. The profile can be used as a template for future experiments on the composition of patches.

## 1. Introduction

Spina bifida (SB) is a congenital neural tube defect that occurs during the early stages of fetal development, which is characterized by the incomplete closure of the neural tube, resulting in a range of spinal cord abnormalities [1]. The neural tube normally forms early in pregnancy, closing around the fourth week of conception [2]. However, in cases of spina bifida, this closure is incomplete, with amniotic fluid entering the fetus and cerebrospinal fluid seeping out of the fetus, resulting in structural defects in the spinal cord that can significantly impact the quality of life and well-being of the patients [3]. The three main types of spina bifida are spina bifida occulta, meningocele, and myelomeningocele, with varying degrees of severity [4]. Spina bifida occulta is the mildest form, involving a small gap in the spine with no visible protrusion [5]. Meningocele involves a sac of cerebrospinal fluid protruding through the opening, while myelomeningocele is the most severe form, where the spinal cord and its protective covering protrude outside the body [6,7,8]. Spina bifida can range from being soft to causing a disability. Symptoms depend on where on the spine the opening is located and how large the gap is. More serious symptoms happen when the spinal cord and nerves are involved. The precise etiologies of spina bifida are complex and multifactorial, involving genetic and environmental factors [9]. Factors such as folic acid deficiency, certain medications, and maternal health conditions may contribute to the occurrence of spina bifida [9,10].

In the United States, about 1500 infants, or 1 in every 2700 births, are born with spina bifida every year [11]. It is a neural tube defect that frequently occurs in families. Spina bifida occurs because of an abnormality in the development of the spinal cord that occurs in the first trimester of pregnancy. Treatment of spina bifida varies based on the severity and type of the condition, and it includes several methods [12]. If the condition is detected early, fetoscopy is a good option for rectifying the problem, having been proven to be safer and more beneficial than traditional surgery [13,14,15]. The latest technology used in the minimally invasive fetoscope prenatal surgery involves deploying a coiled patch through a trocar, expanding the patch at the surgical site, and using tissue sealants or sutures [16]. If the condition is not detected in a timely fashion, the baby will live with the condition with several physical and mental ailments like paralysis and bowel and bladder dysfunction [13]. When necessary, early treatment for spina bifida involves surgery. However, surgery does not always completely restore lost function. Ideally, early screening and diagnosis can reduce the likelihood of damage to the baby.

## 2. Materials and Methods

The Cincinnati Children’s Hospital Medical Center, in cooperation with the Biomedical Engineering Department at the University of Cincinnati, had developed a polymeric patch to protect the defect site and prevent fluid transfer. The patch designed is biocompatible, watertight, self-expanding, and biodegradable. It is a hybrid of poly (L-lactic acid) (PLA) and poly(ε-caprolactone) (PCL) polymers in a 4:1 ratio. PLA degrades quickly, while PCL degrades slowly. Through a series of experiments, it was found that the hybrid patch degrades by an average of ~20% by weight in 16 weeks. Further studies will be conducted in animal models to track degradation beyond 16 weeks. 

The patch has been experimented successfully on rats. One major advantage of the hybrid patch is that it is not necessary to perform a second surgery to remove the patch. The patch is patented (the patch for spina bifida repair is under U.S. Patent No. WO/2018/067811), and the details about the patch were reported in their previous research [16,17,18]. 

The next item on their research agenda was to examine the properties of the hybrid polymer patch in a simulated fetal environment. Inside the womb, the patch faces amniotic fluid (AF) on one side and physiological (cerebrospinal) fluid on the other side. The physiological fluid is chemically represented by phosphate-buffered saline (PBS). Amniotic fluid discarded from fetal surgeries at Cincinnati Children’s Hospital Medical Center was used for experimental purposes with IRB permission (CCHMCIRB#2017-2414). The designed patches are placed in test tubes either soaked in AF or PBS. One of the tasks is to measure patch roughness over a span of time. The reason for measuring roughness is to assess how good the patch is at absorbing nutrients. The higher the degree of roughness is, the stronger the nutrients latch onto the patch, and the speedier the natural skin covers the gap. To measure the roughness of a patch, the patch is subjected to a process, which is destructive. Once the measurement is obtained, the patch is no longer usable. Consequently, how roughness evolves over time cannot be assessed. Despite this acute difficulty in obtaining the requisite data, it is hoped that the assessment over time could be possible in some way. We are addressing this issue in the paper. 

The data we have on hand are destructive. All the roughness measurements come from different patches. The goal is to develop a profile of roughness using this destructive data. This problem is common in pharmaceutical drug testing [19,20,21]. One common research problem in pharmacokinetics is obtaining a profile of how much of a drug remains in the blood. A researcher injects a specific drug at 0 h into a mouse and examines how much drug is left in the bloodstream at several different hours. At any hour of interest, a mouse has to be sacrificed in order to determine the amount of drug in the blood. It is impossible to measure the amount of drug left in the blood for several hours for a single mouse. Pharmaceutical researchers implement the so-called “Sacrifice Design” to collect data [19,20,21]. The classical complete data design where each animal is sampled for analysis once per time point is usually only applicable for larger animals. In the case of rats and mice, where blood sampling is restricted, the batch design or the serial sacrifice design needs to be considered. In serial sacrifice designs, only one sample is taken from each animal. The design involves injecting the drug into 10 animals, for example. At one hour, two animals are sacrificed to measure their drug content. At two hours, another two animals are sacrificed to measure their drug content. This is repeated at 4, 10, 12, and 24 h. We will never have data at all hours of interest for any animal [22,23,24,25]. Our data, in spirit, are similar to the data from the sacrifice design. Monte Carlo methods can be used to recover profile data from the destructive data [25,26]. The Monte Carlo method is a generic name for recovering information from partial data by simulations. In this paper, we introduce an innovative Monte Carlo method to generate profiles of patches from destructive data. Our method is model-based. We pursued two types of Monte Carlo simulations to generate profiles with confidence bands. In one, it was a conditional profile conditioned on information at the 16th week. In the other, it was an unconditional profile covering the entire time span. The details are provided in the Materials and Method Section 2.2.

### 2.1. Experimental Details

Twelve patches were placed in separate test tubes soaked in AF and kept in a shaker. Another twelve patches were placed in separate test tubes soaked in PBS in a shaker. Roughness was measured on an additional three patches as baseline measurements. At four weeks, three patches from AF test tubes were removed and roughness was measured. As has been pointed out, these patches were not reusable. The same process is repeated at eight weeks, twelve weeks, and sixteen weeks. The same is repeated for PBS patches. The data are reproduced in Table 1. Summary statistics are provided in Table 2.

The roughness of a patch rises over time on average, no matter whether the patch was soaked in amniotic fluid or phosphate-buffered saline. Our goal was to build a profile of the roughness of a patch soaked either in AF or PBS at 0, 4, 8, and 12 weeks given *X*_5_. We use a model-oriented endeavor to build the profiles. The method is outlined and implemented in Section 2.2.

### 2.2. Statistical Methods

For any hybrid polymer patch, let *X*_1_ = roughness at zero weeks. After the patch is dipped in AF (or PBS), let *X*_2_ = roughness at four weeks, *X*_3_ = roughness at eight weeks, *X*_4_ = roughness at twelve weeks, and *X*_5_ = roughness at sixteen weeks.

Technically, the vector (*X*_1_, *X*_2_, *X*_3_, *X*_4_, *X*_5_) is not observable in its entirety for any patch. This means, for example, if *X*_1_ is observed for a patch, *X*_2_, *X*_3_, *X*_4_, and *X*_5_ are not observable. In the experiment, three measurements were obtained on each *X_i_* independently from a total of 15 patches. Let (*μ*_1_, *μ*_2_, *μ*_3_, *μ*_4_, *μ*_5_) be the population mean vector of (*X*_1_, *X*_2_, *X*_3_, *X*_4_, *X*_5_). The homogeneity of the means was tested by the ANOVA (analysis of variance) method. The null hypothesis of homogeneity of means was rejected for patches soaked in AF (*p* < 0.001). The homogeneity of population variances was tested by the Bartlett test (*p* = 0.247). The hypothesis of homogeneity of variances was not rejected. An estimate of the common variance was given as 517. The normality and homoskedasticity were checked out to be valid (Wilk–Shapiro test: *p* = 0.705). Similar results hold for patches soaked in PBS (homogeneity of means: *p* = 0.005; normality and homoscedasticity: Wilk–Shapiro Test: *p* = 0.364; homogeneity of variances: Bartlett test: *p* = 0.253). Estimate of the common variance = 490.

Each *X_i_* can be taken to be normally distributed. It is reasonable to assume that (*X*_1_, *X*_2_, *X*_3_, *X*_4_, *X*_5_)~MVN_5_(*μ*, Σ) with mean vector *μ*^T^ = (*μ*_1_, *μ*_2_, *μ*_3_, *μ*_4_, *μ*_5_) = (*μ*^(1)^, *μ*^5^) and dispersion matrix
$$\Sigma =\left(\begin{array}{ccccc} \sigma_{1}^{2} & \rho \sigma_{1}\sigma_{2} & \rho \sigma_{1}\sigma_{3} & \rho \sigma_{1}\sigma_{4} & \rho \sigma_{1}\sigma_{5}\\ \rho \sigma_{2}\sigma_{1} & \sigma_{2}^{2} & \rho \sigma_{2}\sigma_{3} & \rho \sigma_{2}\sigma_{4} & \rho \sigma_{2}\sigma_{5}\\ \rho \sigma_{3}\sigma_{1} & \rho \sigma_{3}\sigma_{2} & \sigma_{3}^{2} & \rho \sigma_{3}\sigma_{4} & \rho \sigma_{3}\sigma_{5}\\ \rho \sigma_{4}\sigma_{1} & \rho \sigma_{4}\sigma_{2} & \rho \sigma_{4}\sigma_{3} & \sigma_{4}^{2} & \rho \sigma_{4}\sigma_{5}\\ \rho \sigma_{5}\sigma_{1} & \rho \sigma_{5}\sigma_{2} & \rho \sigma_{5}\sigma_{3} & \rho \sigma_{5}\sigma_{4} & \sigma_{5}^{2} \end{array}\right)=\left(\begin{array}{cc} \Sigma_{11} & \Sigma_{12}\\ \Sigma_{21} & \Sigma_{22} \end{array}\right),$$
where Σ_11_ is the dispersion matrix of (*X*_1_, *X*_2_, *X*_3_, *X*_4_), and Σ_22_ = ($\sigma_{5}^{2}$). The entity *μ*^(1)^ is the mean vector of (*X*_1_, *X*_2_, *X*_3_, *X*_4_). The way we have partitioned the mean vector and the dispersion matrix is influenced by the following conditional distribution. The acronym MVN stands for multivariate normal distribution.

The *X_i_* s are equi-correlated with common correlation coefficient *ρ*. The dispersion matrix is positive if −1/4 < *ρ* < 1. We have chosen the simple model because it is a reasonable way to build a conditional profile of roughness. We can also handle the conditional probability.

Pr(−a ≤ *X*_1_ − *μ*_1_ ≤ a, −b ≤ *X*_2_ − *μ*_2_ ≤ b, −c ≤ *X*_3_ − *μ*_3_ ≤ c, −d ≤ *X*_4_ − *μ*_4_ ≤ d|*X*_5_), which will be helpful for building a prediction band. Even though we know the conditional distribution of *X*_1_, *X*_2_, *X*_3_, *X*_4_ given *X*_5_, under this model, calculating the conditional probability is extremely difficult. It involves evaluating a four-dimensional integral. However, the distribution can be simulated so that the joint probability can be estimated. This is the gist of the Monte Carlo simulations.

The conditional joint distribution of
$$(X_{1},\;X_{2},\;X_{3},\;X_{4})|X_{5}\sim {\mathrm{MVN}}_{4}(\lambda ,\;\Sigma_{11}-\Sigma_{12}\Sigma_{22}^{-1}\Sigma_{21}),$$
where λ = $\mu^{\left(1\right)}$ + $\Sigma_{12}\Sigma_{22}^{-1}(X_{5}-\mu_{5})$ = $\left(\begin{array}{c} \mu_{1}\\ \mu_{2}\\ \mu_{3}\\ \mu_{4} \end{array}\right)+\rho \left(\begin{array}{c} \sigma_{1}/\sigma_{5}\\ \sigma_{2}/\sigma_{5}\\ \sigma_{3}/\sigma_{5}\\ \sigma_{4}/\sigma_{5} \end{array}\right)$(*X*_5_ − *μ*_5_) and
$$\Sigma_{11}-\Sigma_{12}\Sigma_{22}^{-1}\Sigma_{21}=\Sigma_{11}\;-\rho^{2}\left(\begin{array}{c} \sigma_{1}\\ \sigma_{2}\\ \sigma_{3}\\ \sigma_{4} \end{array}\right)\left(\begin{array}{cccc} \sigma_{1} & \sigma_{2} & \sigma_{3} & \sigma_{4} \end{array}\right).$$

The conditional dispersion matrix is also equi-correlated with correlation *ρ*/(1 + *ρ*). The conditional variance of *X_i_*|*X*_5_ is (1 − *ρ*^2^) × $\sigma_{i}^{2}$. The conditional variance is less now, and the correlation is also less if *ρ* > 0.

Our strategy now works out as follows:Given *X*_5_, simulate the joint distribution of (*X*_1_, *X*_2_, *X*_3_, *X*_4_). This requires knowledge of the conditional mean and conditional dispersion matrix.We need *μ_i_* s, which can be estimated using the individual data on *X_i_* s.We need *σ_i_* s, which can be estimated using the individual data on *X_i_* s.The correlation coefficient *ρ* glues the means, variances, and joint distribution. There was no way we can estimate the correlation coefficient using the marginal data we have. We performed simulations by assuming the value of *ρ* = 0.0 (0.1) 0.9.We conducted Monte Carlo simulations. For each choice of *ρ* and fluid, Steps 1 through 4 were repeated one thousand times. The average of (*X*_1_, *X*_2_, *X*_3_, *X*_4_) s was the desired profile. The 95% band surrounding the mean was built using the following inequality:



$$\begin{array}{c} Pr(-a_{1}\leq X_{1}-\mu_{1}\leq a_{1},\;-a_{2}\leq X_{2}-\mu_{2}\leq a_{2},\;-a_{3}\leq X_{3}-\mu_{3}\leq a_{3},\;-a_{4}\leq X_{4}-\mu_{4}\leq a_{4}|X_{5})\\ \geq \prod_{i=1}^{4}Pr(-a_{i}\leq X_{1}-\mu_{1}\leq a_{i}|X_{5}) \end{array}$$



See Dykstra [27] and Tong [28,29].

Each marginal probability was set at 0.95^0.25 and solved for a. The band was conservative. Simulations were carried out and the results were reported in Section 3.

For the unconditional profile, we took each Xi to be normally distributed. It was reasonable to assume that (*X*_1_, *X*_2_, *X*_3_, *X*_4_, *X*_5_)~MVN_5_(*μ*, Σ) with mean vector *μ*_T_ = (*μ*_1_, *μ*_2_, *μ*_3_, *μ*_4_, *μ*_5_) and dispersion matrix
$$\Sigma =\sigma^{2}\;\left(\begin{array}{ccccc} 1 & \rho & \rho & \rho & \rho \\ \rho & 1 & \rho & \rho & \rho \\ \rho & \rho & 1 & \rho & \rho \\ \rho & \rho & \rho & 1 & \rho \\ \rho & \rho & \rho & \rho & 1 \end{array}\right).$$

Each *X_i_* was assumed to have the same variance. This was justified by the ANOVA procedure carried out in Section 2.2. This model was the classic equi-correlated normal distribution, which means there was the same variance and correlation (*ρ*) between any two *X_i_* and *X_j_*. We took the liberty in assuming equi-correlation. This assumption allowed us build a profile of roughness overtime and a 95% confidence band of the profile. We chose the simple model because this was a reasonable way to build a profile of roughness.

The goal now was to find a number a such that:Pr(−a ≤ *X*_1_ − *μ*_1_ ≤ a,−a ≤ *X*_2_ − *μ*_2_ ≤ a,−a ≤ *X*_3_ − *μ*_3_ ≤ a, −a ≤ *X*_4_ − *μ*_4_ ≤ a, a ≤ *X*_5_ − *μ*_5_ ≤ a) = 0.95.

This probability was a function of the means *μ*_1_, *μ*_2_, *μ*_3_, *μ*_4_, *μ*_5_, *σ*_2_, and *ρ*. We used estimates of means and common variance in the equation. We experimented with several choices of correlation for the band. We chose *ρ* = 0.5 for which the length of each interval 2 × a was minimum. The calculation of the probability was daunting. We resorted to Monte Carlo simulations. The multivariate normal distribution was simulated one thousand times to determine a for our choice of *ρ* [30].

## 3. Results

### 3.1. Conditional Profile

We set *X*_5_ to equal the average of observed *X*_5_. For AF, *X*_5_ = 291 and for PBS, *X*_5_ = 217. The Monte Carlo average profile remained more or less the same across a whole range of *ρ* s. We calculated the average profile at *ρ* = 0.6 for each fluid. 

For amniotic fluid, the band is narrow at 0, 8, and 12 weeks. The band is very wide at 12 weeks. The variances in the marginal data very strongly influence the width of the bands (Figure 1).

### 3.2. Unconditional Profile

The unconditional profiles were stable across time. This was the result of the model we assumed. The vertical width was constant across time. The width was wider for AF than for PBS.

## 4. Discussion and Conclusions

Estimating roughness profile of a patch at 0, 4, 8, and 12 weeks given information on roughness at 16 weeks seemed to be hopeless with the data we had on hand. Our data were destructive in the sense that once roughness was measured on a patch, the patch was no longer useable. We overcame the difficulties by following a model-based approach. We assumed that the roughness measurements on a patch have a multivariate normal distribution with all pairwise correlations equal. By resorting to Monte Carlo simulations [19,20,21], we were able to build the required profile.

The means of roughness were rising over the weeks no matter in what fluid the patches were soaked in. However, there was no clear pattern among standard deviations. The standard deviations of roughness were the largest at 16 weeks for AF and 12 weeks for PBS. Since we were conditioning the profile at 16 weeks, the profile of roughness at the other weeks for AF provided a steady behavior over the weeks. However, the band for PBS at 12 weeks was very wide, reflecting unusual variation present at 12 weeks for PBS.

The unconditional profile was stable over time for both fluids. The stability was due to a constant variance in roughness over time. The assumption of variance was permissible because the ANOVA procedure justified it. According to Figure 2 and Figure 3, the vertical width of the profiles was approximately 100.

This is the first study that develops a biodegradable hybrid patch to cover the gap in the skin of a fetus. The roughness measurements and the profile serve as a template for future research. Future research work would involve creating a hybrid patch with a different composition of polymer patches. The roughness of the new patch can then be compared with our roughness analysis of the hybrid patch.

Inferring the joint behavior of the variables based on marginal behavior is fraught with difficulties. The assumption of equi-correlation seems to be very strong, and its validity is difficult to assess with the limited data available. There are some limitations to our study. There is considerable variation in the marginal data. It seems that no fluid is preferable to the other. Small sample size could be a reason. Sample sizes are typically small in biomedical research. We have informed the core researcher of the need for a reasonable number of samples for a clear understanding of the evolution of roughness over time. In addition, our study is a single-institution study. More trustworthy conclusions could be drawn from a multi-institutional study.

Examining properties of hybrid patches is a natural line of research following the core experiment of covering gaps on the skin of a fetus. In this paper, we laid out how to build a profile using destructive data. The methodology we used is applicable to study other properties. We also plan to explore and improve the methodology to overcome assumptions.

The overarching research goal is to understand how roughness of hybrid patches evolves overtime when the patch is immersed either in AF or PBS. The profile we developed can be used as a template for future experiments on the composition of patches. The experimentalists, as it stands now, understand that amniotic fluid exposure has a higher effect on the surface roughness of the patch.

## Figures and Tables

**Figure 1 bioengineering-11-00249-f001:**
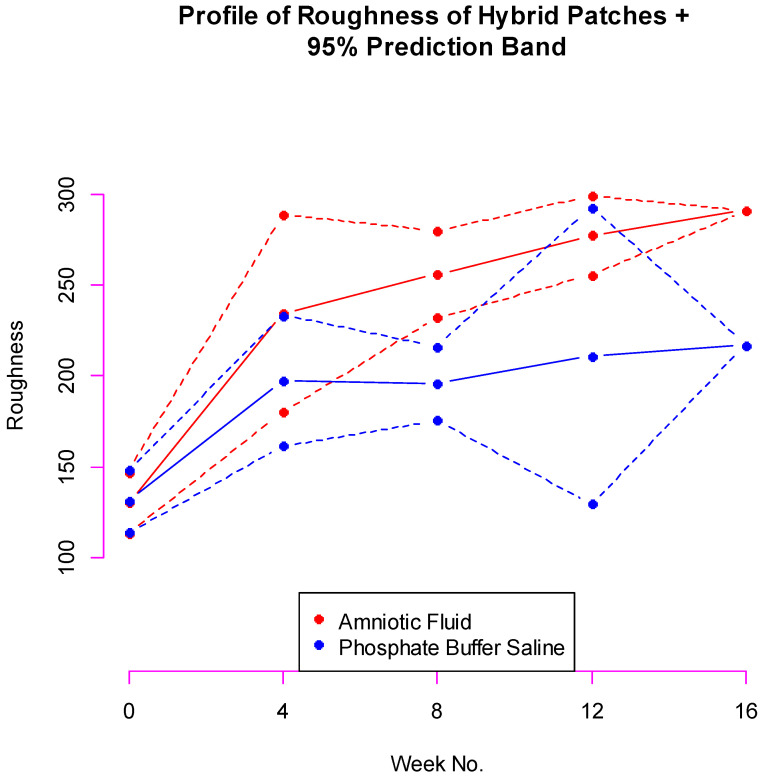
Conditional profile of roughness at zero, four, eight, and twelve weeks given roughness at sixteen weeks + 95% prediction band.

**Figure 2 bioengineering-11-00249-f002:**
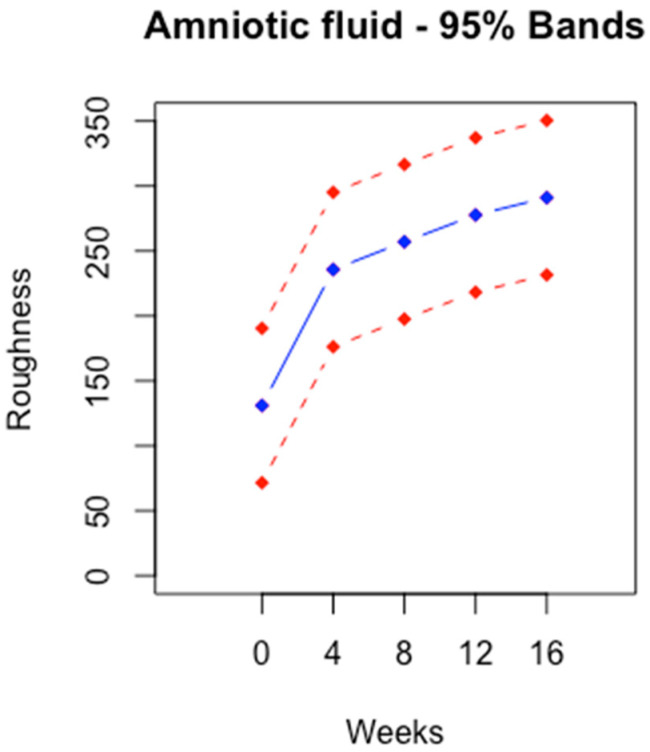
Unconditional profile of roughness at zero, four, eight, twelve and sixteen weeks + 95% prediction band for AF.

**Figure 3 bioengineering-11-00249-f003:**
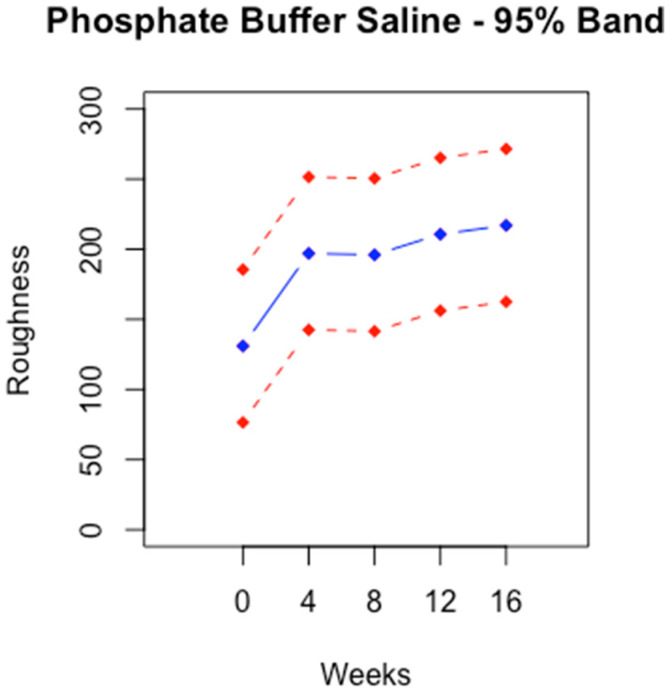
Unconditional profile of roughness at zero, four, eight, twelve and sixteen weeks + 95% prediction band for PBS.

**Table 1 bioengineering-11-00249-t001:** Roughness measurements by fluid and time.

Roughness			
Week	Baseline	AF	PBS
0	139		
0	122		
0	132		
4		223	177
4		267	202
4		217	212
8		245	185
8		269	198
8		257	205
12		265	167
12		283	217
12		285	248
16		306	224
16		247	198
16		320	229

**Table 2 bioengineering-11-00249-t002:** Mean and standard deviation (SD) of roughness by fluid and time.

Week	Baseline		AF		PBS	
	Mean	SD	Mean	SD	Mean	SD
0	131	8.54				
4			235.67	27.3	197	18.03
8			257	12	196	10.15
12			277.67	11.02	210.67	40.87
16			291	38.74	217	16.64

## Data Availability

Reported in Table 1.

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
