# Peer review of "Profile of a Multivariate Observation under Destructive Sampling—A Monte Carlo Approach to a Case of Spina Bifida"

_bioengineering, 2024, doi:10.3390/bioengineering11030249_

Round 1

Reviewer 1 Report

Comments and Suggestions for Authors

The article is interesting and well-written. The introduction is clear and the articles are relevant and recent. Materials and methods are clearly presented, as well as the results. Discussion is sustained by the data presented.

I am attaching further explanation regarding my review:

1.  What is the main question addressed by the research?
The paper aims to test the chemical properties of a patch invented at the University of Cincinnati to cover gaps in the skin over the spine of a growing fetus, characterized by Spina Bifida, a congenital neural tube defect that occurs during stages of fetal development and which is characterized by incomplete closure of the neural tube.

2.  What parts do you consider original or relevant for the field? What specific gap in the field does the paper address?
As clearly presented in the introduction (as per my review, articles considered are relevant and recent; e.g. article n. 20), among the therapies recently developed to treat this issue in an early stage, the use of a patch has been showing promising results. This article aims to test the properties and the durability of the patch.

3.  What does it add to the subject area compared with other published material?
It adds data that clearly show and explain – also thanks to the use of figures and tables, which clearly state the analytical process – the properties of the patch in a controlled, experimental setting. This article builds upon and pushes further the research started on rats, clearly explained in the “Materials and Method” section.

4.  What specific improvements should the authors consider regarding the methodology? What further controls should be considered?
I personally do not suggest any improvement in the methodology, for this specific article.

5.  Please describe how the conclusions are or are not consistent with the evidence and arguments presented. Please also indicate if all main questions posed were addressed and by which specific experiments.
The authors clearly state the process of degradation of their patch and the fact that further research is needed to test its usability (e.g. lines 195-197: “This is the first study on developing a biodegradable hybrid patch to cover the gap in the skin of a fetus. The roughness measurements and the profile service a template for future research”), while still giving valid hypothesis on why no clear result on the validity of the patch has emerged, in this research (lines 200-206).

6.  Are the references appropriate?
The references are appropriate.

7.  Please include any additional comments on the tables and figures and quality of the data.
I don’t think additional data to my review is needed in regards to tables and figures. As stated, I think they are sufficiently informative and efficient.

Author Response

Dear Reviewer,

Thank you so much for your detailed further explanation. Here is my response to your suggestions and comments:

1.The language is streamlined.

  1. The Monte-Carlo procedure is explained in more detail in section materials and methods.

Please review our updated manuscript.

Thanks,

Tian

Reviewer 2 Report

Comments and Suggestions for Authors

Authors present a biodegradable hybrid polymer patch was invented at the University of Cincinnati to cover gaps on the skin over the spinal column of a growing fetus in spina bifida.  A model-based approach with Monte
Carlo simulations was used to estimate the profile over time with a 95% confidence band. The roughness profiles were similar with Amniotic Fluid (AF) or Phosphate Buffer Saline (PBS). The manuscript should be reviewed by individuals who have experience in the bioengineering. As for the clinical part, the main idea is that the patch - if tested for roughness - gets destroyed and unusuable and that is why a model-based approach was developed. I suggest to include similar examples of bioproducts where this type of model can be used. If this is a patent, a patent number should be included.

Comments on the Quality of English Language

Acceptable.

Author Response

Dear Reviewer,

Thank you so much for your detailed further explanation. Here is my response to your suggestions and comments:

  1. Examples from pharmacokinetics are included (lines 84 to 105) and amplified. We also improved our Materials and Methods and Discussion part, please review.
  2. All authors declare that the patch for spina bifida repair is under a U.S. Patent No. WO/2018/067811.

Please review our updated manuscript.

Best,

Tian

Reviewer 3 Report

Comments and Suggestions for Authors

I have some comments:

Materials and Methods section:

The introduction of the Monte Carlo method for recovering profile data from destructive measurements is an innovative approach. However, the details provided in the Materials and Methods section (2.2) are brief. A more thorough explanation of the model-based Monte Carlo method, its assumptions, and its application to the current study would enhance the clarity and reproducibility of the methodology.

The timeline for patch removal and roughness measurement at four, eight, twelve, and sixteen weeks is well-documented in Table 1. However, additional information on why these specific time points were chosen and how they align with the degradation characteristics of the patch would provide context for the readers.

Author Response

Dear Reviewer,

Thank you so much for your detailed further explanation. Here is my response to your suggestions and comments:

  1. We included in detail an explanation of the model-based Monte Carlo method and its application to the current study in the updated manuscript, please review. (line 74 to 104). We added unconditional profile and its use in result and discussion part. Please see our updated manuscript.
  2. We included more background information from published articles in our paper to provide context for readers. Please review.

It was documented that after a patch was laid out on the gap, it took 16 weeks for the natural skin to grow under the patch. Additional three time points were selected to monitor the roughness of a patch.

Best,

Tian

Round 2

Reviewer 2 Report

Comments and Suggestions for Authors

Authors have sufficiently responded to my remarks. 

Comments on the Quality of English Language

Minor changes. 

Reviewer 3 Report

Comments and Suggestions for Authors

The authors have satisfactorily addressed my concerns.